# Identifying and Predicting the Geographical Distribution Patterns of *Oncomelania hupensis*

**DOI:** 10.3390/ijerph16122206

**Published:** 2019-06-21

**Authors:** Yingnan Niu, Rendong Li, Juan Qiu, Xingjian Xu, Duan Huang, Qihui Shao, Ying Cui

**Affiliations:** 1Key Laboratory of Monitoring and Estimate for Environment and Disaster of Hubei Province, Institute of Geodesy and Geophysics, Chinese Academy of Sciences, Wuhan 430077, China; niuyingnan16@mails.ucas.ac.cn (Y.N.); qiujuan@asch.whigg.ac.cn (J.Q.); huangduan@asch.whigg.ac.cn (D.H.); shaoqihui18@mails.ucas.ac.cn (Q.S.) cuiying17@mails.ucas.ac.cn (Y.C.); 2College of Earth and Planetary Sciences, University of Chinese Academy of Sciences, Beijing 100049, China; 3Hubei Provincial Center for Disease Control and Prevention, Wuhan 430079, China; xuxj8412@foxmail.com

**Keywords:** spatiotemporal changes, Maxent, suitable habitats, *Oncomelania hupensis*

## Abstract

Schistosomiasis is a snail-borne parasitic disease endemic to the tropics and subtropics, whose distribution depends on snail prevalence as determined by climatic and environmental factors. Here, dynamic spatial and temporal patterns of *Oncomelania hupensis* distributions were quantified using general statistics, global Moran’s I, and standard deviation ellipses, with Maxent modeling used to predict the distribution of habitat areas suitable for this snail in Gong’an County, a severely affected region of Jianghan Plain, China, based on annual average temperature, humidity of the climate, soil type, normalized difference vegetation index, land use, ditch density, land surface temperature, and digital elevation model variables; each variable’s contribution was tested using the jackknife method. Several key results emerged. First, coverage area of *O. hupensis* had changed little from 2007 to 2012, with some cities, counties, and districts alternately increasing and decreasing, with ditch and bottomland being the main habitat types. Second, although it showed a weak spatial autocorrelation, changing negligibly, there was a significant east–west gradient in the *O. hupensis* habitat area. Third, 21.9% of Gong’an County’s area was at high risk of snail presence; and ditch density, temperature, elevation, and wetting index contributed most to their occurrence. Our findings and methods provide valuable and timely insight for the control, monitoring, and management of schistosomiasis in China.

## 1. Introduction

Schistosomiasis, with freshwater snails as the only intermediate host, is the most devastating infectious disease with an estimated global burden of 3.3 million disability-adjusted life years [1,2]. Not surprisingly, this disease ranks highly in terms of the negative socio-economic effects it has upon afflicted endemic communities [3], especially in the middle and lower reaches of the Yangtze River in southern China, where the disease has been prevalent for at least 2100 years [4,5]. In recent decades, the prevention and control of schistosomiasis in China has made great strides and achievements, with the country now moving closer towards the complete blocking and elimination of schistosomiasis [5]. In 2014, China proposed the ambitious goal of eliminating schistosomiasis by 2025. In addition, the “Healthy China 2030 Planning Outline” and the “13th Five-Year Plan for National Schistosomiasis Prevention and Control” put forward further requirements for prevention and control strategies of schistosomiasis with goals and benchmarks [5]. However, the factors affecting the prevalence and spread of schistosomiasis—in particular, the wide distribution of snails in the epidemic zones of schistosomiasis, the complex environment, the serious spread of local snails, and the epidemic-promoting factors of most schistosomiasis endemic areas—still persist, as does the risk of schistosomiasis outbreaks recurring in some areas [5,6]. Therefore, achieving the scheduled target of complete elimination of schistosomiasis by 2025 still faces many challenges [7].

Sound knowledge of the distribution of intermediate hosts for schistosomiasis is of paramount importance for the effective design, implementation, and evaluation of programs to control the disease [8]. Even when relying on cheap and effective medications, a better understanding of the local dynamics of snail populations remains crucial for large-scale drug management under a global strategy, as recognized by the World Health Organization, which is useful in those periods when re-infection is very low for reducing the likelihood of re-emergence [3]. However, the sustainability of such a control strategy is questionable because re-infection can be rapid even after implementing deworming initiatives, and the focus on morbidity control has since shifted transmission control and local elimination [3]. Hence, there is a stronger focus on snails and transmission sites, along with primary prevention tailored to specific socio-ecological systems [3,9,10].

It is acknowledged that the presence of *O. hupensis* snails will depend on specific climatic and environmental conditions, which governs the distribution of schistosomiasis [11]. Many variables can jointly affect the growth and reproduction of snails due to complex physical geography, including meteorological factors [12,13], such as temperature and precipitation, as well as environmental factors [14,15,16], such as vegetation, topography, land use, land surface temperature, and water sources.

The use of spatial information technology mainly depends on the application and development of geographic information systems (GIS), global positioning systems (GPS), and remote sensing (RS) techniques. Such technology has powerful spatial data management and analysis functions that offer unique advantages for handling location-based geographical data. These spatial tool functions can be used effectively to study the relationships between various natural and social factors and human health in a specific geographical area, revealing spatiotemporal patterns in health or disease and how their geographically relevant factors are related [17]. For example, spatial information technology has been widely applied to various global diseases, namely malaria [18], dengue fever [19], and West Nile disease [20]. When investigating an infectious disease epidemic, GIS is mainly concerned with four aspects: making a disease distribution map, determining the spatial distribution characteristics of the disease or its media, addressing and discussing risk factors, and formulating time-space predictions [17]. In this context, RS provides real-time, continuous, high-efficiency large-area remote sensing data in the form of vegetation index, soil moisture, and surface temperature variables. It is thus a cost-effective way to monitor the ecological environment in terms of its atmosphere, water and soil, and urban pollution and heat-island effects. For these reasons, in recent years RS has been widely used to study disease-related environmental factors [2,3,17].

The availability of such detailed environmental data, together with inexpensive and powerful computers, has fueled a rapid increase in the predictive modeling of species environmental requirements and their geographical distributions [21]. Maximum entropy (Maxent) modeling technology can greatly enhance our understanding of the intermediate host snails of schistosomiasis; on this basis, snails may be controlled and managed in a more effective way and this knowledge could suggest new ways to tackle the problem of schistosomiasis in disease-prone regions [3]. Therefore, in this study, we first analyzed the temporal and spatial changes of snail habitat areas in the Jianghan Plain, an area heavily afflicted by schistosomiasis in China. Then, we applied Maxent modeling to determine the potential habitats of *O. hupensis* in Gong’an County, the main ditch habitat area for this snail species, as an example to analyze the potential distribution areas of *O. hupensis*.

## 2. Materials and Methods

### 2.1. Study Area

Jianghan Plain, one of the most severely affected endemic areas of schistosomiasis in China, is in the central-south of Hubei Province (Figure 1). Forming the main part of the middle and lower reaches of the Yangtze River, the Jianghan Plain covers approximately 30,000 km^2^, spanning eight counties and municipalities, including Jingzhou District, Shashi District, Jiangling County, Gong’an County, Jianli County, Shishou City, Honghu City, Songzi City, as well as three cities under direct provincial management: Xiantao City, Qianjiang City, and Tianmen City. Since the Jianghan Plain lies in a subtropical monsoon climate zone, it has a warm and humid climate, with abundant light and heat resources. These environmental characteristics provide ideal conditions for the freshwater snail *O. hupensis*, which is the only intermediate host of *Schistosoma japonicum*.

### 2.2. Data Acquisition

#### 2.2.1. *Oncomelania hupensis* Survey Data

The *O. hupensis* data set used came primarily from annual field surveys in the fall, and it consisted of two parts: (1) known *O. hupensis* habitat coverage areas of the administrative villages from 2007 to 2012, and (2) line-transect data, both in the Jianghan Plain. This data set was obtained from the Institute of Schistosomiasis Control, based at the Hubei Provincial Center for Disease Control and Prevention. 

Because of the availability and accuracy of land surface temperature and land use data, we used the *O. hupensis* distribution map at the administrative village scale in 2010, combined with the linear transects’ data of *O. hupensis*, an objective determination of the snail’s distribution points was conducted along the linear transects. Specifically, in the administrative village without *O. hupensis*, no points were selected; if the administrative village had a large area of *O. hupensis*, more points were drawn along the linear transects, but if the contrary, fewer points were drawn. According to these rules, 468 *O. hupensis* distribution points were obtained (Figure 1).

#### 2.2.2. Environmental Data

To predict and analyze the suitable habitat area of *O. hupensis* in Gong’an County, among the heavily affected areas of Jianghan Plain, the following nine environmental variables were considered: annual average temperature (Bio_1), annual average precipitation (Bio_2), humidity of the climate (Bio_3), soil type (Bio_4), normalized difference vegetation index (Bio_5), land use (Bio_6), ditch density (Bio_7), land surface temperature (Bio_8), and digital elevation model (Bio_9) (Table 1).

Annual average temperature (Bio_1) and precipitation (Bio_2), humidity of the climate (Bio_3), soil type (Bio_4), land use (Bio_6), and digital elevation model (Bio_9), all came from the Data Center for Resources and Environmental Sciences, Chinese Academy of Sciences (RESDC) (http://www.resdc.cn), were re-sampled to 1-km spatial resolution. The 30-m spatial resolution data on normalized difference vegetation index (Bio_5) and land surface temperature (Bio_8) were calculated and retrieved from Landsat TM5 imagery, acquired on 16 September 2010, after which ArcGIS software was used to re-sample both variables to a 1-km resolution. In addition, six GF-1 images (2-m spatial resolution) covering Gong’an County were acquired on 7 November and 19 November 2013, and three multispectral images (16-m spatial resolution) with the same coverage were acquired on 7 December 2013. Together, these were used to identify and locate ditches on the landscape of Gong’an County, via the screen vectorization method; 1-km spatial resolution ditch density (Bio_7) raster data for the study area were then calculated in ArcGIS. 

Based on the above 1-km environmental factor raster data, the vector layer covering Gong’an County was applied to extract the raster environmental factor layers. Then, Pearson’s correlation coefficient was used to analyze associations between the above factors, of which annual average precipitation (Bio_2) was eventually excluded due to its high correlation with humidity of the climate (Bio_3).

### 2.3. Methodology

#### 2.3.1. Distribution Analysis

##### Global Moran’s I

Moran’s I is a commonly used measure of global spatial autocorrelation [22,23]. Here, it was applied to detect whether or not there was spatial autocorrelation in the coverage area of *O. hupensis* habitat from 2007 to 2012. Spatial autocorrelation can be defined as the property of random variables taking values, at pairs of locations a certain distance apart, that are more similar (i.e., positive autocorrelation) or less similar (i.e., negative autocorrelation) than expected for randomly associated pairs of observations. Autocorrelation is a very general property of all variables observed along time series or across geographic space [24]. For quantitative or continuous variables, Moran’s I is the most commonly used coefficient in univariate autocorrelation analyses, and was calculated here as follows [25]:(1)$$I=(\frac{n}{s})[\frac{\sum_{i}\sum_{j}\left(y_{i}-y_{mean})(y_{j}-y_{mean}\right)w_{ij}}{\sum_{i}{\left(y_{i}-y_{mean}\right)}^{2}}]\;s=\sum_{i=1}^{n}\sum_{j=1}^{n}w_{ij}$$
where *n* is the number of villages; *y_i_* and *y_j_* are the coverage areas(m^2^) of *O. hupensis* habitat in villages *i* and *j*; *y_mean_* is the average coverage areas(m^2^) of *O. hupensis*; and *w_ij_* is spatial weight matrix between village *i* and village *j*. In this paper, we use the method of inverse distance to determine the spatial relationship between administrative villages; that is, compared with the remote administrative villages, the nearer neighboring administrative villages have a greater impact on the calculation of the target administrative village. Moran’s I usually varies between −1 (indicating perfect dispersion) and +1.0 (perfect correlation), with a zero value indicating a random spatial pattern. For statistical hypothesis testing, Moran’s I values can be transformed to *Z*-scores for which values greater than 1.96 or less than −1.96 indicate spatial autocorrelation that is significant at the 5% alpha level.

##### Standard Deviation Ellipse

The standard deviation ellipse (SDE), first introduced in 1925 [26], is a spatial statistical method that quantitatively describes the spatial distribution characteristics of geographic features of interest [27]. The SDE depicts directional bias present in a set of points; more details about this methodology are available in reference [26]. Here, we sought to explore whether *O. hupensis* distributions from 2007 to 2012 stayed the same or changed over time. To do this, village-based *O. hupensis* habitat centered Cartesian coordinates (X,Y) were first weighted by *O. hupensis* habitat area, with total area and eccentricity of each SDE used to quantitatively compare the habitat of *O. hupensis* in each year. The center corresponded to the mean distribution center of each year’s *O. hupensis* habitat. A small SDE would suggest the distribution of *O. hupensis* is fairly compact relative to a larger SDE; in short, the distribution of *O. hupensis* habitat is more concentrated. Eccentricity provides an indication of the polarity of the distribution within the ellipse; a more polar distribution will tend to yield a lower eccentricity value. Conversely, an eccentricity value closer to 1 would suggest the points are more uniformly distributed around the mean center; hence, any directionality would not be obvious [26].

#### 2.3.2. Niche-Based Models

Maximum entropy method is a general-purpose machine learning method with a simple and precise mathematical formulation, which has many features that make it well suited for species distribution modeling, especially for those with presence-data only (refer to [21] for more details). Here, to model the suitability of habitat for *O. hupensis*, we used MaxEnt v.3.4.1 (http://biodiversityinformatics.amnh.org/open_source/maxent/). We generated models that randomly assigned 75% of *O. hupensis* occurrences as the training data, with the remaining 25% used as the test (validation) data, with the replicated run type set to bootstrap. The default settings of Maxent were adhered to for the regularization multiplier (value of 1) and maximum number of background points (value of 10,000), and we used the logistic output format conditioned on the environmental variables in each grid cell, with values ranging from 0 (= unsuitable habitat) to 1 (= optimal habitat) [28]. Model performance was expressed as the area under (the receiver operator characteristic) curve (i.e., AUC) supported by sensitivity and specificity [3]. An AUC value of 0.5 indicates that the model’s predictive ability is no better than that of a random model, while AUC values >0.75 are considered in the “best” model category [29]. A jackknife procedure, implemented in Maxent, was applied to quantify the explanatory power of each environmental variable.

## 3. Results

### 3.1. Spatiotemporal Variation of Oncomelania hupensis Habitat Areas

We analyzed the total area of *O. hupensis* habitat and the main *O. hupensis* habitat types in various counties, cities, and districts from 2007 to 2012. As Figure 2 shows, the habitat area of *O. hupensis* in Xiantao City (more than 70 million m^2^), which tended to decrease at first (2007–2010) but then the increase (2010–2012), was the largest. For Jianli County, the trend was of increasing (2007–2008), then decreasing (2008–2009), and not changing (2009–2012). The habitat area of *O. hupensis* in Shishou City (decreased in 2007–2008, unchanged during 2008–2011, and decreased in 2011–2012) was slightly lower than that in Jianli County, yet slightly larger than that in Honghu City (decreased during 2007–2009, unchanged during 2009–2012). In addition, compared with those four areas, the *O. hupensis* habitat areas were all smaller in Gong’an County (increased in 2007–2008, but decreased during 2008–2010, unchanged during 2010–2012), Jiangling County (increased in 2007–2008, then decreased during 2008–2012), Qianjiang City (decreased during 2007–2012) and Songzi City (decreased during 2007–2012). From 2007 to 2012, the habitat areas of *O. hupensis* in Shashi District, Tianmen City and Jingzhou District were all less than 10 million m^2^.

Moreover, to analyze the main *O. hupensis* habitat types in each region, we carried out a statistical analysis on their respective areas. As Figure 3 shows, the *O. hupensis* snails are mainly distributed in ditch and bottomland habitats. Different from bottomland, the main *O. hupensis* habitat type in Xiantao City, Songzi City, Honghu City, Shishou City and Jianli County, ditch was also the dominant habitat type of *O. hupensis* in Jiangling County, Qianjiang City, Shashi District, Gong’an County and Jingzhou District, with Gong’an County having the largest area of ditch *O. hupensis*. However, the pattern for Tianmen City differed from the others, with bottomland and ditch being the main distribution types, respectively, from 2007 to 2009 and 2010 to 2012.

### 3.2. Spatial Correlation in Coverage Areas (m^2^) of O. hupensis Habitat

It can be seen from Table 2 that, from 2007 through 2012, the Moran’s I index remained positive and close to a value of 0, thus indicating spatial autocorrelation of *O. hupensis* habitat areas of administrative villages in the Jianghan Plain was consistently weak, with a distribution mode close to random. That is to say, any aggregation—the administrative villages with large distribution areas of *O. hupensis* are adjacent to each other, or the administrative villages with small distribution areas of *O. hupensis* are adjacent to each other—of *O. hupensis*’ areas was not obvious. Nevertheless, Moran’s I did vary slightly from year to year.

### 3.3. Directional Analysis of O. hupensis Habitat Areas

These results are summarized in Figure 4 and Table 3. Firstly, as the former shows, the distribution centers of *O. hupensis* areas in Jianghan Plain were located in Jianli County. Generally, these distribution centers changed little from 2007 to 2012, yet a slight shift was evident, and the distribution center in 2007 was clearly to the north, while that over later years stayed mostly the same. Secondly, Table 3 shows that the areas of the standard deviation ellipse (SDE) followed a trend of increasing at first, then decreasing, then increasing again, and finally decreasing; this indicated the distribution of *O. hupensis* in Jianghan Plain underwent an initial diffusion, then aggregation, then diffusion, and finally aggregation. Finally, both the distribution of SDE (Figure 4) and the eccentricity variation (Table 3) revealed a clear east–west gradient in *O. hupensis* distributions from 2007 through 2012 in our study region.

### 3.4. Potential Distribution Region of O. hupensis

The potential geographic distribution of *O. hupensis* snails was produced with a full set of occurrence localities by applying the Maxent modeling method. The training AUC and test AUC values were 0.922 and 0.8971, respectively (Figure 5); hence, indicative of good prediction accuracy. We used the ‘natural breaks’ classification method to differentiate the potential distribution of *O. hupensis* into four risk levels, and also calculated the areas corresponding to these *O. hupensis* habitat types: namely “unsuitable, negligible risk” (0–0.13, 574 km^2^), “poorly suitable, low-risk” (0.13–0.33, 401 km^2^), “suitable, medium-risk” (0.33–0.52, 795 km^2^), and “highly suitable, high-risk” (0.52–0.84, 496 km^2^ ) (Figure 6). As Figure 6 shows, with the exception of small parts of northern, central, and southern Gong’an County, most of its areas are suitable as *O. hupensis* habitat. Among them, the high-risk areas are mainly distributed in some small grouped areas in the central and eastern parts of Gong’an County (encircled by thin black lines).

The habitat distribution of host snails was influenced by several environmental factors, based on the jackknife algorithm (Figure 7). Clearly, Bio_7 (ditch density) contributed much more than any other factor examined; however, Bio_1 (temperature), Bio_9 (digital elevation model), and Bio_3 (humidity of the climate) also had a considerable impact for predicting the *O. hupensis*’ habitat.

In addition, response curves were used to determine how each environmental variable influenced the model. The four most influential factors were then used to discern the general impact of environmental factors on snail occurrence (Figure 8). It can be seen from the response curve of the ditch density (Bio_7, Figure 8a), that as the density of the ditch increases, the probability of *O. hupensis* occurrence increases. When the ditch density is ca. 6.6 km per square kilometer, the occurrence probability peaks at 75%, but with greater ditch density the likelihood decreases to 0.18. 

The response curve of the annual average temperature (Bio_1, Figure 8b) and humidity of the climate (Bio_3, Figure 8d) indicated that within their respective ranges for the study region, the probability of *O. hupensis* occurrence was positively correlated with these two factors. Apart from this, as the elevation increased, the occurrence probability also increased, reaching its maximum at ca. 30 m, after which the probability of *O. hupensis* occurrence decreased (Figure 8c).

## 4. Discussion

### 4.1. Temporal and Spatial Variation in Oncomelania hupensis Area Distribution

*O. hupensis*, by functioning as the intermediate host where parasites can quickly multiply, plays a major role in spreading schistosomes to humans, so *O. hupensis* control strategies are considered a priority for reducing transmission rates of this disease [30,31]. Although China has implemented various measures to control the snail populations of *O. hupensis* [32,33], their total distribution area in various counties and cities in the Jianghan Plain has changed little. The spatiotemporal variation in *O. hupensis* distributions is mainly affected by various interacting natural and social factors as well as the snail’s life history [34]. According to available data on the epidemic situation of schistosomiasis in the People’s Republic of China over the past years, the distribution area of *O. hupensis* has declined from 14.8 billion m^2^ in the early period of the country’s founding to 3.56 billion m^2^ in 2015. This *O. hupensis* area declined significantly between 1955 and 1980, but from 1980 through 1988 it tended to increase, while between 1989 and 2015 the national *O. hupensis* area generally fluctuated [34,35].

The conditions providing optimal habitat for *O. hupensis* are mainly a mix of water, sunlight, air, weeds, and temperature attributes [36]. In this respect, weeds around a ditch, siltation of the ditch, and long-term water accumulation and disturbance are needed for *O. hupensis* snails to breed [36,37]. Key water attributes influencing the distribution of *O. hupensis* in bottomland has been linked to flooding time, soil water content, and groundwater level [38,39], of which the latter likely has the greatest impact on snail density. Research has also demonstrated that the bottomland distribution of *O. hupensis* is significantly related to the groundwater level; when this sits at ca. 32 cm, the snail density was found to be largest. Therefore, changes in local water levels can shape micro- and macro-environmental features of *O. hupensis* habitat that affect the life activities and distribution of these snails, thereby influencing the spread of schistosomiasis [38]. Based on our findings, both ditches and bottomlands should be targeted as key areas for controlling *O. hupensis* reproduction and abundance.

Although several measures—namely, building forests and *O. hupensis*-resistant facilities, hardening channels used by *O. hupensis*, and traditional drug-induced mortality—were implemented to eliminate the species during the 2007–2012 period, the response of *O. hupensis* snails in different places tends to vary [40,41,42,43,44]. This may partly explain the changes in the spatial autocorrelation index of *O. hupensis* areas in administrative villages across years. For example, two villages each with a large *O. hupensis* distribution area may be adjacent to each other, yet also adjacent to a third village having a small *O. hupensis* distribution area, thus generating a signal of randomness. Although the area encompassing *O. hupensis* underwent discernible change, administrative village-scaled *O. hupensis* distributions were basically unchanged, so their standard deviation ellipse yielded an approximately similar level of coverage.

### 4.2. The O. hupensis Habitat Suitability and Its Response to Environmental Variables

Maxent models do not predict the actual limits of a species’ range but can identify localities with similar conditions for their respective occurrence [3]. The Maxent model has been applied successfully for prediction of many species distributions [21,45], including snails [3,9,46]. According to model evaluation criteria [29], the Maxent model based on our eight environmental factors provided robust prediction of the *O. hupensis* snail distributions in Gong’an County. Importantly, we found the habitat suitability of *O. hupensis* varied from location to location, differing in its sensitivity to different environmental factors. Previous study has revealed that the normalized difference water index (NDWI), a proxy for surface water, was more significant and consistent in its seasonal distribution of suitable habitats of schistosomiasis intermediate host snails in the Ndumo area, KwaZulu-Natal Province, South Africa [3]. However, in our study, we found that ditch density, annual average temperature, elevation, and humidity of the climate were the four most influential factors suggested by our *O. hupensis* modeling, with ditch density making the strongest contribution. Weeds and sludge in primary ditches are common in our study region, where pollution from garbage waste and manure is substantial, all of which generates a very slow water flow; in the long run, the *O. hupensis* snails could easily reproduce in these weeds and their fecal contamination can swiftly re-infect the local aquatic environment [36,47]. Hence, we recommend the local water conservancy department should regularly clean ditches to maintain sufficient water flow, so that it along with other schistosomiasis control measures can better control the growth of *O. hupensis* populations [36,47]. Although some ditch-hardening projects have been implemented, the density of hardened ditches with *O. hupensis* is still very low [32,37,47,48], while the density of unhardened ditch segments connected to them remains comparatively high; hence, *O. hupensis* can still propagate in water flowing through already hardened ditches [48]. Therefore, when designing and building a hydraulic engineering project for *O. hupensis* control, its funding should be arranged in an integrated manner, so as to reach the goal of controlling and eliminating *O. hupensis* populations. 

Furthermore, annual average temperature was also a critical factor in our modeling of *O. hupensis* habitat suitability. This was expected, since temperature has been shown in other studies to affect the distribution of this snail [12,34,49,50,51]. Zhou et al. [50] have argued the potential for a northward shift of *O. hupensis* will increase with global warming. For this species, the suitable temperature for mating is 15–20 °C, being unsuitable at 30 °C or more and 10 °C or less, while the most suitable temperature for its spawning is 20–25 °C [51]. The development of *O. hupensis* eggs and their incubation time are mainly determined by water and temperature. When the latter is 13–23 °C, incubation time is shortened with greater temperatures, after which ambient temperature significantly affects the life of *O. hupensis* [51].

Understanding the distribution elevation of *O. hupensis* is crucial for understanding this snail’s ecological characteristics [52]. In our study, the probability of *O. hupensis* occurrence varied with elevation, being greatest at ca. 27 m. A previous study [53] on the elevation of the *O. hupensis* distribution in Wuhan, which has similar terrain to Gong’an County in the Jianghan Plain, found an average elevation of 17.72 ± 8.64 m for existing *O. hupensis* distribution points. Therefore, the two results are consistent with each other.

Some work [38,39] has found *O. hupensis* distributions to be closely associated with soil moisture, but no studies have yet reported a relationship between *O. hupensis* distribution and humidity of the climate. In this paper, we showed that within a certain range, as the air humidity index increases the probability of occurrence of *O. hupensis* also increases. This relationship between the air humidity and the distribution of *O. hupensis* thus warrants further investigation.

### 4.3. Implications for O. hupensis Control Programs

Analyzing the temporal and spatial variation characteristics of *O. hupensis* snails, as well as their potential suitable areas at the micro-geographic scale, may assist in carrying out *O. hupensis* control measures [34]. The main techniques used in China to kill snails of this species include drug-killing them, and ecological *O. hupensis*-killing methods. Survival of *O. hupensis* is augmented in wet lakes, beaches, grassy beaches, or similar environments, with ecological *O**. hupensis*-killing technology now attracting more and more attention because it alters and destroys the snail’s habitat, in way that is mostly safe, efficient, economical, and environmentally friendly [48]. For example, by transforming dry land waterlogged fields, waterlogged low-yield fields, and adjusting the agricultural planting structure, the crop-planting operation mode could be shifted from the industrial structure of water to drought, leaving the snail’s environment deteriorated [48]. In addition, through the implementation of ditch hardening, the vegetation environment has been completely changed, so that the snails lose their survival and breeding sites, enabling a long-lasting effect of killing snails to be achieved [48]. Although the presence of *O. hupensis* is necessary to determine a potential area of schistosomiasis transmission, it alone is not sufficient to qualify as a transmission site. If the *O. hupensis* snails are not infected in the area and there is no human-to-water contact, transmission to humans does not occur there. Therefore, while useful, the prediction of *O. hupensis* habitat suitability may not suffice for developing novel schistosomiasis control, monitoring, or management schemes [3].

## 5. Conclusions

This study had two main objectives: analyzing spatiotemporal variation characteristics of the *O. hupensis* distribution area in Jianghan Plain as well as taking its Gong’an County—a severely afflicted endemic schistosomiasis area—as an example to predict *O. hupensis* habitat suitability based on remotely sensed environmental data. Our results showed that the total distribution area of *O. hupensis* in various counties and cities in Jianghan Plain changed little over a 6-year period. Importantly, *O. hupensis* were mainly distributed in ditches and bottomland, habitat types that deserve more attention. Although some measures to eliminate *O. hupensis* snails were in place, non-uniform response to these efforts generated some autocorrelation and directional distributions that were mostly the same from 2007 to 2012. Additionally, Maxent modeling, with a good accuracy, revealed that most areas in Gong’an County provide high habitat suitability for *O. hupensis*. According to jackknife testing of variables’ contribution to modeling this *O. hupensis* habitat suitability distribution, ditch density, temperature, elevation, and humidity of the climate had the strongest relative influences. These findings and methods may offer insight into controlling, monitoring, and management of schistosomiasis in China.

## Figures and Tables

**Figure 1 ijerph-16-02206-f001:**
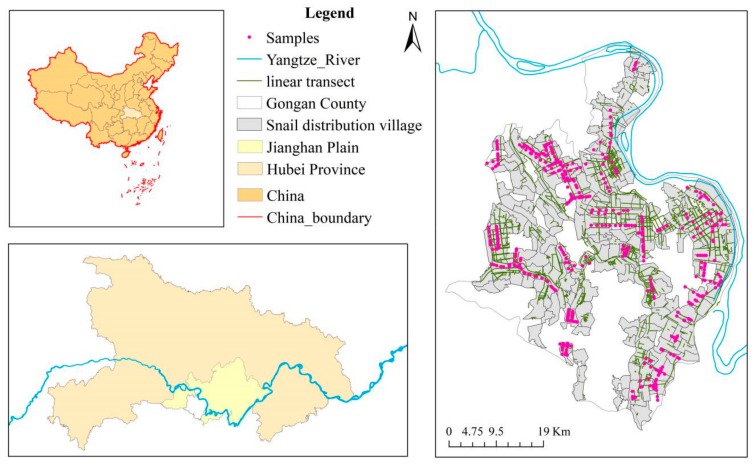
Location of the study area in China and *O. hupensis* snail samples.

**Figure 2 ijerph-16-02206-f002:**
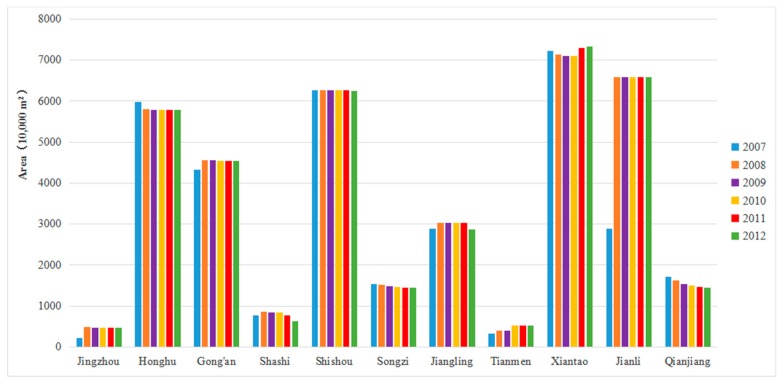
Gross *O. hupensis* snail habitat area.

**Figure 3 ijerph-16-02206-f003:**
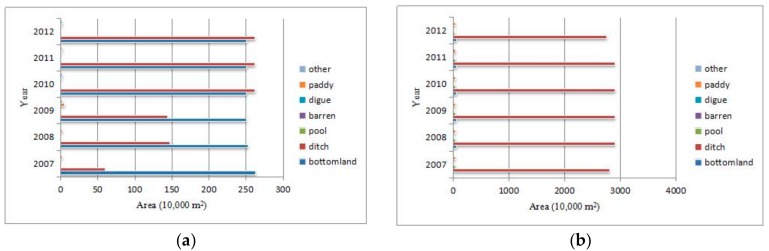
Coverage areas of *O. hupensis* in different land use types. (**a**) Tianmen City; (**b**) Jiangling County; (**c**) Qianjiang City; (**d**) Xiantao City; (**e**) Shashi District; (**f**) Songzi City; (**g**) Gong’an County; (**h**) Honghu City; (**i**) Jingzhou District; (**j**) Shishou City; (**k**) Jianli County.

**Figure 4 ijerph-16-02206-f004:**
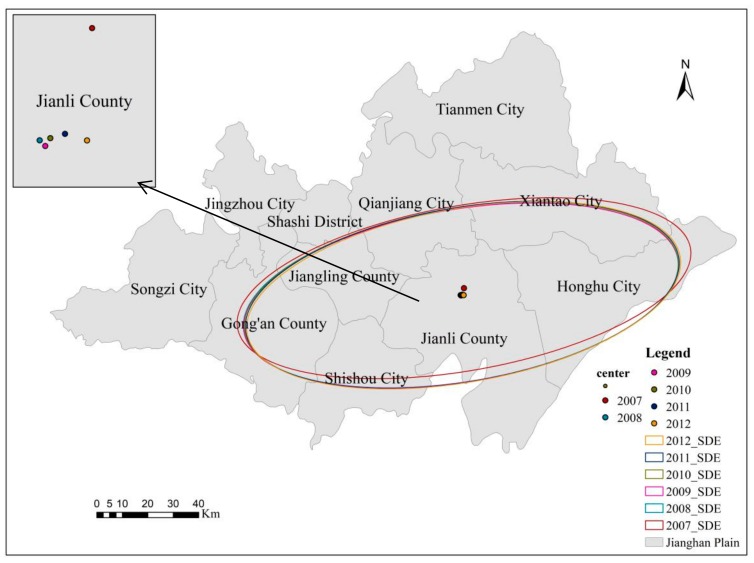
The SDE (standard deviation ellipse) of *O. hupensis* snail areas of Jianghan Plain, China, from 2007 to 2012.

**Figure 5 ijerph-16-02206-f005:**
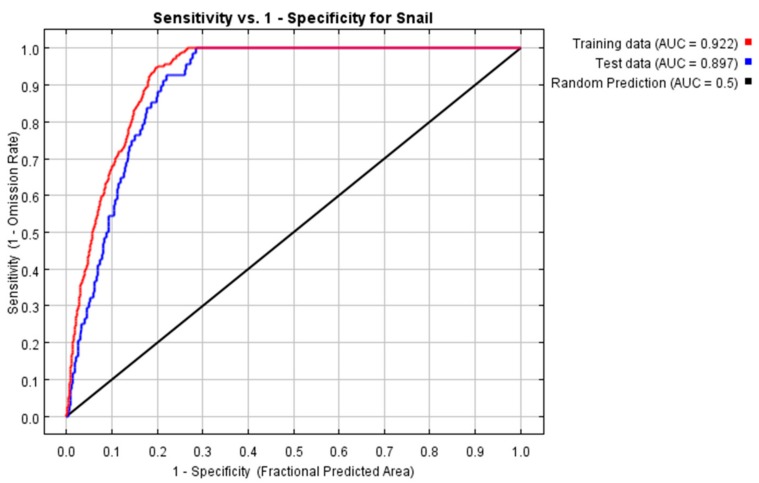
Values from the AUC (area under curve) testing.

**Figure 6 ijerph-16-02206-f006:**
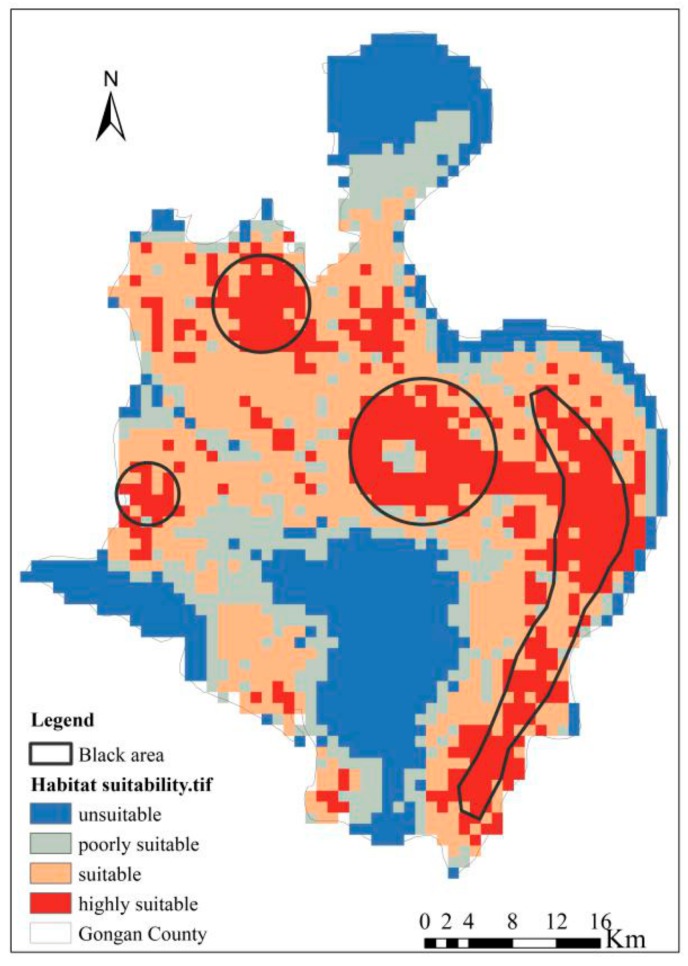
Predicted potential geographic distributions for *O. hupensis* snail using all occurrence records and environmental variables, by applying Maxent modeling at a 1-km resolution.

**Figure 7 ijerph-16-02206-f007:**
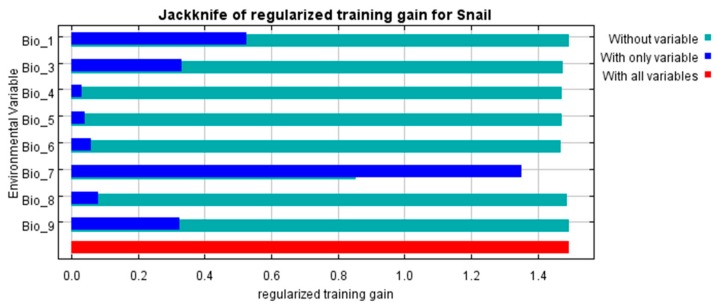
Results of jackknife testing of each variables’ contribution to modeling the *O. hupensis* habitat suitability distribution. Note: Bio_1, the annual average temperature; Bio_3, humidity of the climate; Bio_4, soil type; Bio_5, normalized difference vegetation index; Bio_6, land use; Bio_7, ditch density; Bio_8, land surface temperature; Bio_9, digital elevation model.

**Figure 8 ijerph-16-02206-f008:**
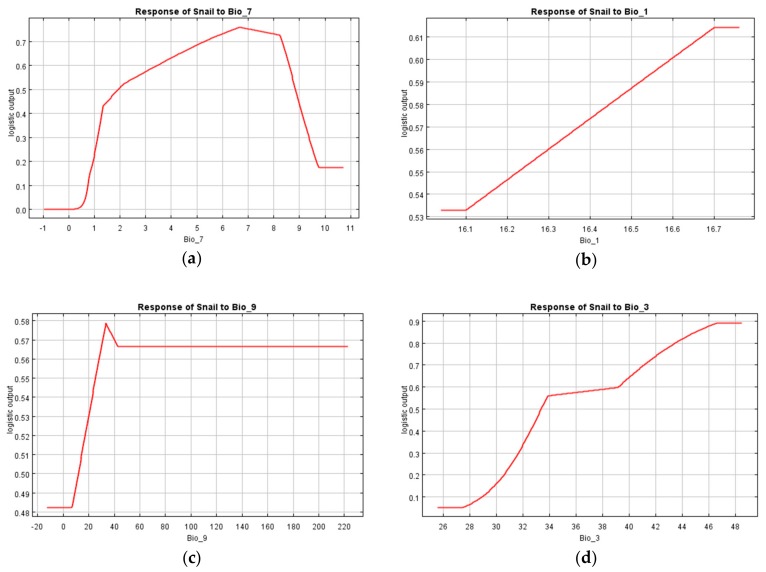
Response curves of the four major environment variables. The x-axis and y-axis represent the value of a factor and the probability of *O. hupensis* occurrence, respectively. Note: Bio_1, annual average temperature; Bio_3, humidity of the climate; Bio_7, ditch density; Bio_9, digital elevation model. (**a**) ditch density; (**b**) temperature; (**c**) digital elevation model; (**d**) humidity of the climate.

**Table 1 ijerph-16-02206-t001:** Environmental variables used in the study.

Code	Environmental Variables	Data Type	Data Source	Spatial Resolution
Bio_1	annual average temperature	Continuous	http://www.resdc.cn	500 m
Bio_2	annual average precipitation	Continuous		
Bio_3	humidity of the climate	Continuous		
Bio_4	soil type	Categorical		1 km
Bio_5	normalized difference vegetation index	Continuous	retrieved from Landsat TM5 imagery	30 m
Bio_6	land use	Categorical	http://www.resdc.cn	1 km
Bio_7	ditch density	Continuous	extracted from GF-1 images	2 m
Bio_8	land surface temperature	Continuous	retrieved from Landsat TM5 imagery	30 m
Bio_9	digital elevation model	Continuous	http://www.resdc.cn	

**Table 2 ijerph-16-02206-t002:** Moran’s I index of *O**. hupensis* snail habitat coverage areas (m^2^) at the administrative village scale.

Year	Moran’s I	*Z*-Score	*p*-Value
2007	0.048296	10.94	<0.001
2008	0.039563	8.17	<0.001
2009	0.03975	8.17	<0.001
2010	0.039821	8.20	<0.001
2011	0.039538	8.19	<0.001
2012	0.040216	8.17	<0.001

**Table 3 ijerph-16-02206-t003:** Measures of yearly SDE (standard deviation ellipse).

Year	Area (km^2^)	Center_X	Center_Y	Eccentricity
2007	9219.52	757159.13	3204851.46	0.360
2008	9245.36	755873.93	3202085.97	0.394
2009	9229.85	756010.59	3201945.43	0.394
2010	9318.00	756134.84	3202139.83	0.398
2011	9307.29	756495.94	3202244.58	0.398
2012	9277.80	757036.95	3202082.25	0.396

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
