# Peer review of "Identifying and Predicting the Geographical Distribution Patterns of Oncomelania hupensis"

_ijerph, 2019, doi:10.3390/ijerph16122206_

Round 1

Reviewer 1 Report

This article refers to the identification and prediction of the geographical distribution patterns of Oncomelania hupensis in the Jianghan plain area, one of the most intense endemic regions of schistosomiasis in China.

The authors exploited a series of data that refer, on the one hand, to annual surveys on the 2007-2012 period and on environmental and climatological data, which were adopted on a single scale. They also exploited methods and tools of geospatial technologies (GIS, GPS, and RS) in order to analyze their data as well as several related bibliographic references that are helpful for the methods and technologies they used. However, their general methodological approach presents, in my point of view, ambiguities and shortcomings but also indications of directivity in the results of the research.

1.       Specifically, the authors while designating the Jianghan plain area (figure 1) as their study area and collecting and analyzing data for the whole area, apply their model (Maxent model) especially in the Gong'an region because as they claim this area is the "main ditch habitat area for Oncomelania hupensis" (lines 88-92). That is, they give weight to one of the 9 environmental parameters that have subsequently been declared to affect the spatial distribution of Oncomelania hupensis (lines 127-128). With all due respect to my fellow authors, I want to say that perhaps there is an intention to direct their results in order to arrive at the desired conclusions.

2.       They report the use of a map (Oncomelania hupensis distribution map), but they do not show it in the article (line 115) and analyze unclearly the way to adopt Oncomelania hupensis distribution points (lines 115-120). I believe that it needs there a further explanation.

3.       They do not explain at all why Pearson's correlation coefficient does not show an association of the environmental average annual precipitation (Bio_2) with the other parameters. This parameter, however, has absolute relevance to the key enhancer of snail habitat that is responsible for the transmission of the disease being tested.

Nevertheless, if one treats the article with a broad spirit, he will find that the use of Niche-based models and the analysis made in the discussion section show that the authors have a thorough knowledge of the scientific objects this research is being dealt with and seriously examines the geo-epidemic behavior of the reference disease. So my personal opinion is leaning towards the publication of this important research, as long as the questions raised in this review are answered.

Best regards

Author Response

Thanks a lot for your comments concerning our manuscript entitled “Identifying and Predicting the Geographical Distribution Patterns of Oncomelania hupensis”(Manuscript ID ijerph-492723). Those comments are very helpful for revising and improving our paper. We have studied comments carefully according to your suggestions. All changes made to the manuscript are highlighted so that they may be easily identified. Please find our detailed response attached.

Reviewer 2 Report

The authors used a variety of analytical methods in the paper. But the use of these methods was not well justified: why are these, not other, methods are used? Are they really necessary? What’s the rationale behind each method?

The paper reports little new information. Many of the findings, derived from seemingly sophisticated analyses, are common sense. For example, the authors claim that a major finding was that Oncomelania hupensis were mainly distributed in ditches and bottomlands, and these areas should be targeted as key areas for controlling. For local people and most researchers in this field, this is common sense.

The authors correctly pointed out that “the spatiotemporal variation in Oncomelania hupensis distributions is mainly affected by various interacting natural and social factors as well as the Oncomelania hupensis’ life history”, but their analysis failed to include any social factor

There are many grammatical and stylistic errors in the paper. The writing needs to be seriously revised and improved before the paper can be accepted for publication in an English journal.   

Author Response

(The authors gave the same response as above.)

Reviewer 3 Report

The manuscript by Dr. Niu and colleagues presents a geospatial analysis of the distribution of the snail species Oncomelania hupensis, the invertebrate host species of Schistosoma japonicum, in the Jianghan Plain region of China, an area that is still severely affected by schistosomiasis. First, the authors use survey data about the spatial occurrence of the target snail species to study statistical features like spatial autocorrelation and directionals biases in the species distribution. Then, based on a set of environmental covariates, they perform a species distribution modeling exercise using Maxent for Gong'an county (one of the administrative units included in their study area), with good training and validation results. I think this paper deals with an interesting topic, seems to be based on sound methodological premises, and presents some results that are potentially useful for the management of the snail population in the target region, with larger implications for integrative control strategies for the fight against schistosomiasis. The manuscript is generally well written and clearly organized, and should be appealing to a wide readership. All this being said, I have some technical comments that the author may want to address while revising their work.

General comments

- I did not completely understand how the survey data from different sources (village-based surveys, linear transects) were blended together to produce the distribution map used for further spatial analyses. Is there a way to show the original data and their final elaboration in a figure accompanying section 2.2.1? Also, it would be interesting to have some additional pieces of information about the survey data: for instance, are there any estimates of snail population density and/or of prevalence of infection with S. japonicum?

- I think the authors should provide some justifications for their selection of environmental covariates. For instance, several indicators seem to refer (directly or not) to `average' environmental conditions, while temporal variability appears to be somewhat underrepresented. In other, arguably different, geographic contexts (e.g. sub-Saharan Africa), seasonality of temperature, precipitation and waterbody ephemerality are typically important factors for the ecology of the snail species that host Schistosoma haematobium or S. mansoni. I am not quite familiar with the target species and the environmental conditions of the study area, but I wonder if e.g. seasonal flooding might play a role here, also given the reported prominence of bottomland habitats in the study region.

- Related to the previous point, I also think it is important that the authors provide a technical description of each environmental covariate they use. As an example, the definition of `wetting index', or the difference between `annual average' and `land surface' temperature indicators may be not familiar to some readers.

- Concerning the results of the species distribution modeling exercise, it would be interesting to compare the results presented in this work with those described in other studies using a similar approach in different geographical areas (hence, perhaps for different host species of schistosomiasis), some of which are already referenced in the manuscript.

Minor comments

- l.46 is the occurrence of schistosomiasis in those areas indeed `epidemic'? 

- l.61 `a sole intermediate host snails' in unclear (and grammatically awkward)

- l.72 nothing wrong with those examples, but I would argue that nowadays it is kind of difficult to name one disease for which some sort of `spatial information technology' has not been used

- l.105 perhaps it is worth stressing that Schistosoma japonicum is the parasite being referred to (other schistosomes have other intermediate host species)

- l.107 the figure caption is a bit misleading: the main panel is the study area; the inset represents its location within Hubei Province (not generically within China)

- l.124 why has the analysis been restricted to Gong'an county alone?

- l.154 I would perhaps stress again that Moran's I measures spatial autocorrelation

- l.158 how has the spatial weight matrix been defined? It would be interesting to know whether it is just based on geographical proximity, or whether hydrological connectivity has also been factored in somehow

- l.236 is there a way to test the level of significance of the year-to-year variations of the SDE?

- l.381 I would suggest to better explain the meaning of `ecological killing technology'

- although the manuscript is generally well written, there are quite a few typos that need to be taken care of, as shown by the following (likely incomplete) list: 

l.16 unnecessary comma at the end of the line

l.35 transmission disease?

l.45 snails is misspelled

l.109 Oncomelania hupensis must be typeset in italics (here and everywhere)

l.149 coverage is misspelled

l.150 period missing after habitat

l.158 coverage is misspelled

l.197 Oncomelania hupensis must be typeset in italics (here and everywhere)

l.208 period missing before `In the Shash District'

l.265 snails is misspelled

l.306 Oncomelania hupensis must be typeset in italics (here and everywhere)

l.311 Oncomelania hupensis must be typeset in italics (here and everywhere)

l.316 findings is misspelled

Author Response

(The authors gave the same response as above.)

Round 2

Reviewer 3 Report

I think the authors have done a fair job of addressing the comments raised during the first round of review. One thing that is still missing, though, is a brief yet technical description of each environmental covariate selected for analysis. These descriptions might usefully be placed in a table (but a revision of section 2.2.2 would do as well, of course). As an example, it seems that in the revised version of the manuscript the authors still feel the need to characterize their `wetting index' in layman terms at the end of the manuscript (p.14, l.409). I would advise against these casual characterizations in favor of a more rigorous description of the environmental variables where they are first introduced in the manuscript.

Author Response

Dear reviewer,
Thanks a lot for your suggestions concerning our manuscript entitled “Identifying and Predicting the Geographical Distribution Patterns of Oncomelania hupensis”(Manuscript ID ijerph-492723).
We have added the descriptions of environment variables in Table 1 at the end of the section 2.2.2. Besides that, we have replaced wetting index with humidity of the climate everywhere in the article, and all changes have been colored in green.

1. In line 20, we replaced wetting index with humidity of the climate.
2. In line 129, we replaced wetting index with humidity of the climate.
3. In line 131, we added (Table 1) .
4. In line 132-135, the data descriptions have changed to ( Annual average temperature (Bio_1) and precipitation (Bio_2), humidity of the climate (Bio_3), soil type (Bio_4), land use (Bio_6), and digital elevation model (Bio_9), all came from the Data Center for Resources and Environmental Sciences, Chinese Academy of Sciences (RESDC) (http://www.resdc.cn), were re-sampled to 1-km spacial resolution.)
5. In line 135, The 30-m2 space resolution data were changed to The 30-m spacial resolution data.
6. In line 138, 1-km2 were changed to 1-km.
7. In line 140, 16-m2 spacial resolution were changed to 16-m spacial resolution.
8. In line 142, 1-km2 spacial resolution were changed to 1-km spacial resolution.
9. In line 144, 1-km2 were changed to 1-km.
10. In line 147-148, wetting index has been replaced with humidity of the climate.
11. We added Table 1 at the end of the section 2.2.2.
12. In line 240,247, 1 has been replaced with 2.
13. In line 250,254, 263, 2 has been replaced with 3.
14. In line 275, along the Yangtze River were changed to Gong’an County.
15. In line 286, 295, 303, 308, 310, 360, 394, 430, wetting index has been replaced with humidity of the climate.

Once again, we want to express our thanks sincerely for your comments and suggestions.